Applied and Environmental Science

mSystems®

# Size Shapes the Active Microbiome of the Methanogenic Granules, Corroborating a Biofilm Life Cycle

Anna Christine Trego,[a] Sarah O'Sullivan,[a] Christopher Quince,[b] Simon Mills,[a] Umer Zeeshan Ijaz,[c] Gavin Collins[a,c]

[a]Microbiology, Ryan Institute and School of Natural Sciences, National University of Ireland Galway, Galway, Ireland
[b]Warwick Medical School, University of Warwick, Coventry, United Kingdom
[c]Water Engineering Group, School of Engineering, University of Glasgow, Glasgow, United Kingdom

**ABSTRACT** Methanogenic archaea are key players in cycling organic matter in nature but also in engineered waste treatment systems, where they generate methane, which can be used as a renewable energy source. In such systems in the built environment, complex methanogenic consortia are known to aggregate into highly organized, spherical granular biofilms comprising the interdependent microbial trophic groups mediating the successive stages of the anaerobic digestion (AD) process. This study separated methanogenic granules into a range of discrete size fractions, hypothesizing different biofilm growth stages, and separately supplied each with specific substrates to stimulate the activity of key AD trophic groups, including syntrophic acid oxidizers and methanogens. Rates of specific methanogenic activity were measured, and amplicon sequencing of 16S rRNA gene transcripts was used to resolve phylotranscriptomes across the series of size fractions. Increased rates of methane production were observed in each of the size fractions when hydrogen was supplied as the substrate compared with those of volatile fatty acids (acetate, propionate, and butyrate). This was connected to a shift toward hydrogenotrophic methanogenesis dominated by *Methanobacterium* and *Methanolinea*. Interestingly, the specific active microbiomes measured in this way indicated that size was significantly more important than substrate in driving the structure of the active community in granules. Multivariate integration studywise discriminant analysis identified 56 genera shaping changes in the active community across both substrate and size. Half of those were found to be upregulated in the medium-sized granules, which were also the most active and potentially of the most important size, or life stage, for precision management of AD systems.

**IMPORTANCE** Biological wastewater conversion processes collectively constitute one of the single biggest worldwide applications of microbial communities. There is an obvious requirement, therefore, to study the microbial systems central to the success of such technologies. Methanogenic granules, in particular, are architecturally fascinating biofilms that facilitate highly organized cooperation within the metabolic network of the anaerobic digestion (AD) process and, thus, are especially intriguing model systems for microbial ecology. This study, in a way not previously reported, provoked syntrophic and methanogenic activity and the structure of the microbial community, using specific substrates targeting the key trophic groups in AD. Unexpectedly, granule size more strongly than substrate shaped the active portion of the microbial community. Importantly, the findings suggest the size, or age, of granules inherently shapes the active microbiome linked to a life cycle. This provides exciting insights into the function of, and the potential for additional modeling of biofilm development in, methanogenic granules.

**KEYWORDS** community assembly, anaerobic digestion, biofilm, methanogenic granules, microbial community assembly

Address correspondence to Umer Zeeshan Ijaz, umer.ijaz@glasgow.ac.uk, or Gavin Collins, gavin.collins@nuigalway.ie.

Specific active microbiome assays indicate that size, more than substrate, shapes the microbial community of methanogenic granules, corroborating a biofilm life-cycle in anaerobic digesters

**B**iofilm formation is an evolutionarily ancient process, providing microorganisms with architectural rigidity, protection from environmental stresses, and a matrix for organized microbial community assembly (1). Biofilms can establish emerging features not always possible outside the biofilm environment, such as nutrient gradients across trophic groups and aerobic or anaerobic microniches (2). Classical models of formation and growth of attached biofilms are widely accepted (1, 3), but biofilm development and community assembly by complex microbiomes is still of increasing interest (4–8). A challenge for such studies is in identifying highly replicated, complex, and diverse communities that experience identical environmental conditions, especially at scales relevant to cells (5).

To this end, granular biofilms, usually found in engineered wastewater treatment plants (Fig. 1), are emerging as ideal "playgrounds" to test fundamental concepts in microbial ecology (5, 8, 9). Granules are self-immobilized, usually spherical, biofilm aggregates, which, in all identified cases, contain highly complex and diverse microbial communities comprised of multiple trophic groups working in coordination or competing with one another (10–13). Granules have recently been used in creative ways to study (i) the role of species sorting during microbial community assembly (8); (ii) strain-level diversity and whether communities can be grouped into types at various taxonomic levels (5); and (iii) the role of quorum sensing during both community assembly and biofilm disintegration (4). We recently used methanogenic granules to study diversity across a wide range of biofilm sizes, finding that the microbiomes of differently sized granules differ significantly, and, interestingly, that in the granules of the largest sizes, diversity converges toward a core microbiome of methanogenic archaea (14). Moreover, we found that size is likely a strong marker of granule growth, where small granules may be considered young and large granules old (15).

The microbial consortia of anaerobic granules span several trophic groups, including hydrolyzers, fermenters, homoacetogens, syntrophic $H_2$-producing acetogens, sulfate reducers, and methanogens, underpinning the mineralization of complex organic matter to methane through the anaerobic digestion (AD) process (16, 17). The linchpin of such consortia is the methanogens, holding obvious importance for waste treatment engineers as the producers of valuable methane (18) but also of persistent interest to evolutionary biologists as some of the most ancient contemporaneous organisms on Earth (19). The activity of methanogenic archaea is routinely measured using specific methanogenic activity (SMA) assays against key AD substrates and intermediates (20, 21). The assays typically monitor headspace pressure in sealed incubations of biomass and a specific substrate (e.g., $H_2/CO_2$, acetate, propionate, or butyrate). This study combined rRNA sequencing of the active (cDNA-based) community in SMA assays of methanogenic granules with two main objectives, to (i) unravel the ecophysiology of granules fed with key, specific substrates and (ii) determine whether the structure of the active community is influenced by granule size and as part of an apparent granule life cycle.

## RESULTS

**Methanogenic activity and targeted metabolomic profiles.** Granules, sourced from a full-scale anaerobic digester treating potato-processing wastewater in the Netherlands, were size separated into nine size fractions (Fig. 1) and separately used to inoculate a range of SMA assays. These assays capture a snapshot of methanogenic activity against several specific methanogenic substrates: acetate, propionate, butyrate, and hydrogen. SMA assays revealed gradients in methanogenic activity according to size. Volatile fatty acid (VFA)-stimulated activity was significantly higher ($P = 0.001$ [***]) in medium-sized granules (fractions E to G) than in smaller or larger granules (Fig. 2). However, hydrogenotrophic methanogenic activity was generally higher overall across all of the granule sizes (Fig. 2). VFA profiles from the respective sampling points indicated not all of the acetate, propionate, or butyrate was fully consumed during the tests. Additionally, more complex VFA compounds, especially isobutyric and isovaleric

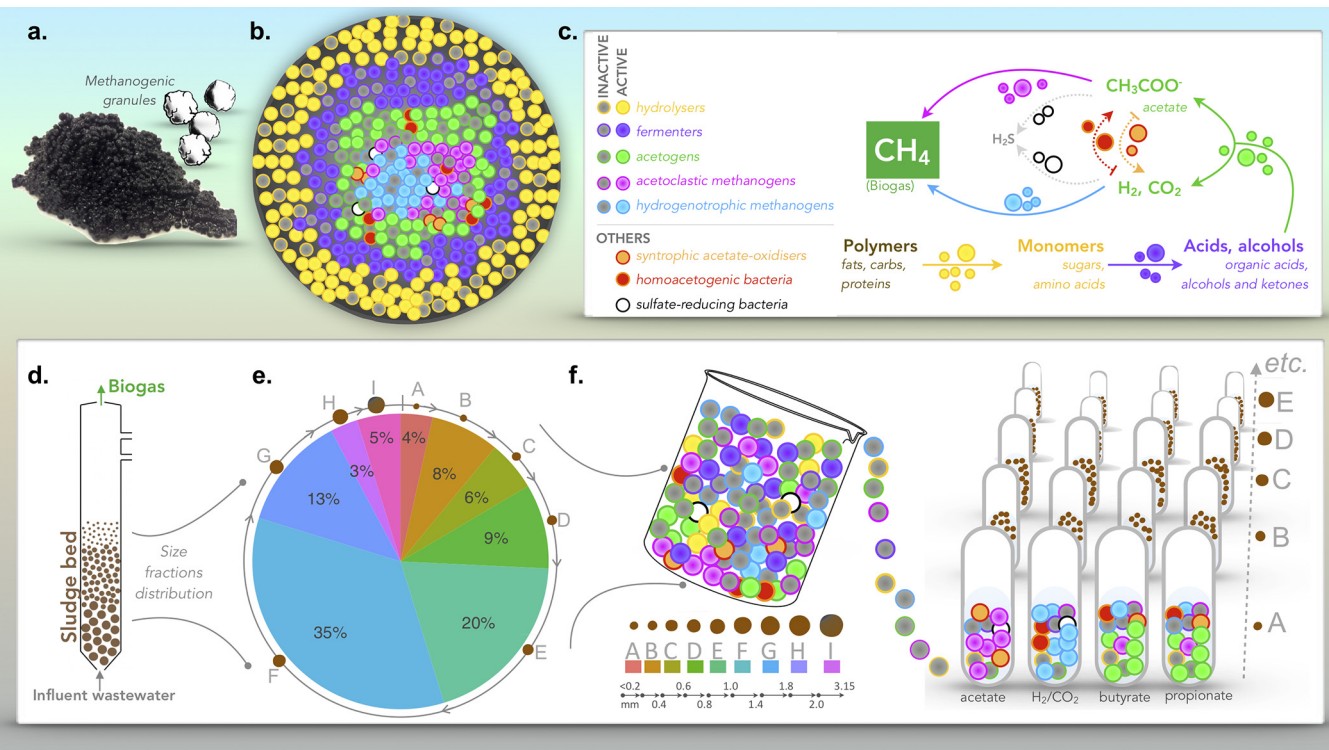

**FIG 1** Schematic. (a) Anaerobic sludge and representation of individual granules. (b and c) Diversity (b) and theoretical distribution (c) of the main microbial trophic groups in the anaerobic digestion pathway. (d) Laboratory-scale anaerobic digester with a granular sludge bed. (e) Proportional distribution of the nine granule size fractions recovered from the anaerobic sludge used for this study. (f) Size fraction endpoints and SMA assays set up with each of the size fractions, indicating the four substrates used, acetate, hydrogen ($H_2$/$CO_2$), butyrate, and propionate. The trophic groups hypothesized to be predominant in the assays are shown in panel f according to the scheme in panel c.

acids, were detected, although at low concentrations. The profiles showed significant differences ($P = 0.001$ [***]) according to granule size.

**Structure of the active microbial community across size and substrate.** Highly significant ($P = 0.001$) differences were observed in the structure of the active portion of the microbial community across granule size, regardless of substrate (Fig. 3), with fewer significant differences between the substrate-supplied and no-added-substrate control tests. Feeding with $H_2$/$CO_2$ resulted in significantly reduced richness and evenness in the active community across all granule sizes (Fig. 3). However, VFA substrates produced strong shifts in alpha diversity only in smaller granules (fractions A and B). Environmental filtering, or phylogenetic alpha diversity, revealed strong patterns across granule size but weaker patterns according to substrate (Fig. 4). The net relatedness index (NRI) decreased with increasing granule size, indicating the phylogeny of the active community was more clustered in smaller granules and more evenly spread in larger granules. Phylogenetically clustered communities usually result from strong selective processes, either environmental selection or biotic, niche-based processes.

Beta diversity analysis and permutational analysis of variance (PERMANOVA) suggested that granule size influenced the active microbiome more strongly than substrate (Fig. 5). In fact, regardless of substrate, differences according to size were always highly significant ($P = 0.001$). Substrate choice did, indeed, shift the structure of the active microbiome across all tests, but it did so less significantly. For example, according to the Bray-Curtis distance metric for the acetate-fed tests, 56% ($P = 0.001$ [***]) of community structure shifts were attributed to granule size, while only 2% ($P = 0.028$ [*]) could be attributed to acetate. The addition of hydrogen resulted in the most pronounced community response and shifted the community similarly across each of the size fractions.

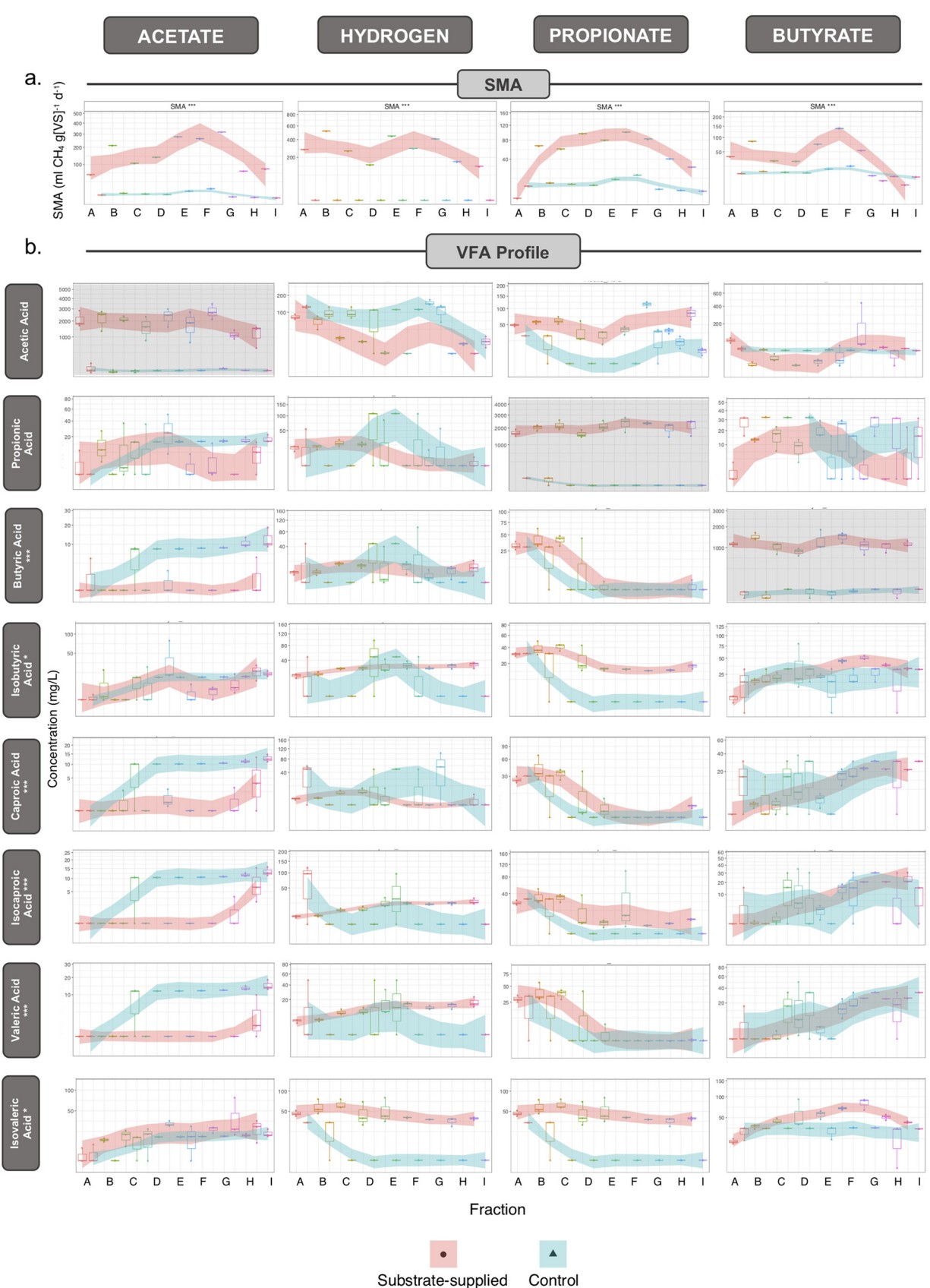

**FIG 2** (a) SMA (*n* = 3) from each of the size fractions (A to I) against acetate, hydrogen (H₂/CO₂), propionate, and butyrate. (b) VFA profiles (mg VFA liter⁻¹), resolving acetic, propionic, butyric, isobutyric, caproic, isocaproic, valeric, and isovaleric acids, of each SMA assay sampled during

Analysis of the top 25 most abundant operational taxonomic units (OTUs) showed clear shifts from the total (DNA-based) to the active (cDNA-based) communities (Fig. 6). While OTUs representing, for example, *Hyd-24-12* and *Draconibacteriaceae* were highly abundant in the total microbiome, they were far less prominent in the active community. Meanwhile, many of the OTUs, such as those representing *Eubacterium acidamino-philum*, which were relatively rare in the total microbiome, comprised a majority of the active community in the SMA assays. With $H_2/CO_2$ as the substrate, a hydrogenotrophic community made up of more *Methanobacterium* than that observed in the VFA-supplied tests was favored. Additionally, the relative abundance of active-community components showed strong gradients across granule size.

Substrate addition stimulated the activity of various clades, differing in sizes, and usually including various branches of the *Euryarchaeota* but also including groups of the *Proteobacteria*, among others (see Fig. S1 in the supplemental material). Finally, multivariate integration (MINT) studywise discriminant analysis identified 56 specific genera responsible for changes in the active community structure across both substrate and size (Fig. S2). The majority of these discriminant OTUs were fermentative bacteria, but several methanogenic archaea were also linked to particular substrates. For example, methanobacterium was upregulated when the community was supplied with hydrogen as the sole substrate.

## DISCUSSION

**Microbiome gradients across size and substrate: size matters.** Previous studies had identified gradients in the composition of the total community of methanogenic granules according to biofilm size (14). Here, we observed similar gradients in the active microbiome, in which small granules contained more diverse, and rich, active communities than larger granules. This was observed in the substrate-supplied tests as well as the no-added-substrate control tests, which represented a sort of background activity, and suggests diversity gradients across size, with biofilm age, toward a core, active microbiome. The background activity appeared to be more variable in small granules (fractions A and B) than in larger granules; that this pattern was unidirectional suggests an underlying ecological process at play in which smaller granules are more highly influenced by environmental conditions. Furthermore, considering the hypothesis originally proposed by Díaz et al. (22), and corroborated by subsequent research (15), that small granules can be considered young and larger granules old, such gradients point to mechanisms of biofilm development and continued community assembly over a biofilm lifespan.

Unsurprisingly, methanogenic archaea (the phylum *Euryarchaeota*) dominated the active community. The active community in granules of all sizes yielded high relative proportions of methanogens, both in the control tests and the substrate-supplied tests. However, the substrates did provoke the upregulation of the *Euryarchaeota*. Hydrogen addition had a strong effect across all of the sizes, upregulating euryarchaeotal clades of *Methanobacteriaceae* but also bacterial clades of *Clostridia* and *Deltaproteobacteria*. Acetate, propionate, and butyrate even more widely affected the active microbiomes, stimulating specific clades of *Euryarchaeota* but also a wide range of other bacteria, suggesting other pathways of substrate utilization were activated and that clades were competing for substrates, a conclusion also supported by the diversity of VFA produced. The upregulation of specific clades also appeared to be dependent on granule size. Acetate, propionate, and butyrate all appeared to stimulate the upregulation of *Euryarchaeota* in smaller granules, suggesting that size matters for structure and function. This may be due to something in the architecture of the young, small biofilms,

**FIG 2** Legend (Continued)
exponential biogas production. (VFA profiles and DNA/RNA extracts were from the same samples.) Shaded regions (using LOESS smoothing) track the shifts of the substrate-supplied (pink) and no-added-substrate control (blue) values. The *y* axis for all profiles is square root transformed to place emphasis on smaller values. Plots shaded entirely in gray indicate the concentrations of the substrate supplied and, at that point, not yet consumed.

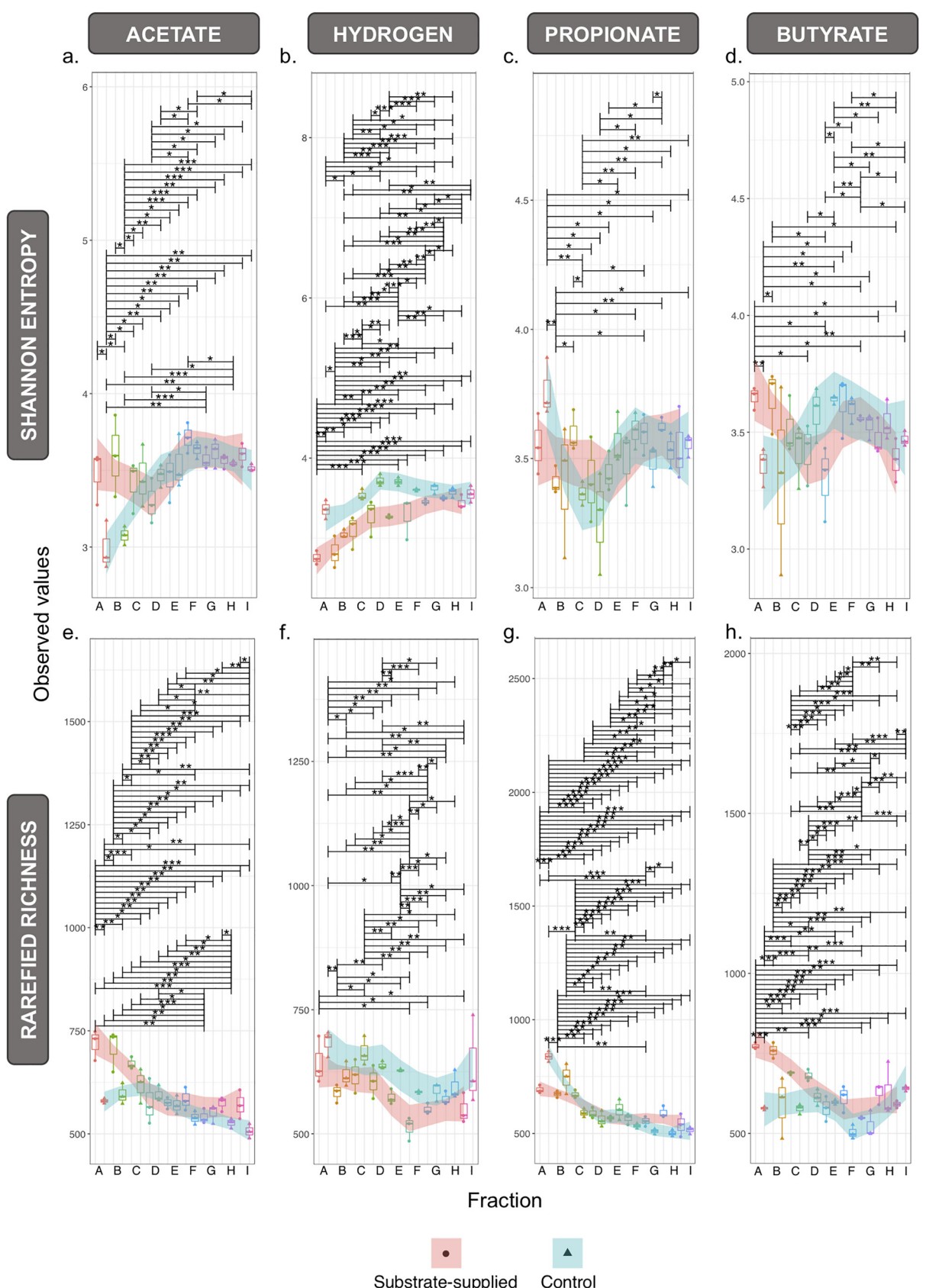

**FIG 3** Microbial alpha diversity according to variances in rRNA sequences from cDNA in samples ($n = 3$) from across each of the nine size fractions, A to I, supplied with specific substrates (acetate, $H_2/CO_2$, propionate, and butyrate) or the respective no-added-substrate control.

the makeup of the extracellular polymeric substance (EPS) matrix, or the propensity for substrate diffusion.

**Size, more than substrate, drives diversity of the active microbiome.** PERMANOVA additionally suggested that granule size (or life stage) was a more significant force driving diversity gradients than substrate supplementation. That substrate plays a significant role in shaping the microbiome in AD communities (23, 24), as well as in soil (25, 26) and marine environments (27–29), is well documented. Nonetheless, this was overshadowed by the influence of granule size on community dynamics, perhaps representing biofilm development stages. Across each substrate tested, and against every metric, granule size was a stronger driver of the active community structure. Moreover, size appeared to influence, to some degree, the impact of substrate on community structure.

**Community response to $H_2$ is stronger than that to soluble, VFA substrates.** Of all the substrates, the provision of $H_2/CO_2$ produced the strongest observable shifts in the active community structure. The shift from predominantly heterotrophic to autotrophic methanogenesis is more extreme than the more modest pathway shifts between different VFA. Indeed, acetogens, such as *Syntrophomonas*, are capable of oxidizing a range of short-chain VFA (30). Such metabolic versatility, however, is rare among methanogenic archaea. Only species belonging to the *Methanosarcina* are considered metabolically flexible, capable of both chemoautotrophy and chemoheterotrophy (31). Conversely, a more diverse collection of methanogens is autotrophs capable of using only $H_2$ and $CO_2$ (32). Irrespective of size, $H_2/CO_2$ significantly increased the relative abundances of *Methanobacterium* and *Methanolinea* but also resulted in size-related shifts in bacterial taxa, including *Syntrophomonas*, *Oligosphaera*, and *Geobacter*. While *Syntrophomonas* and *Geobacter* shifted from dominance in larger granules to smaller granules, *Oligosphaera* became more important in larger ones. These groups do not directly generate methane and most likely are syntrophic partners in the AD process.

Moreover, $H_2/CO_2$ supplementation resulted in the highest rates of methanogenesis in the SMA tests, regardless of granule size. Previous reports suggested stressed AD systems shift toward hydrogenotrophic methanogenesis (33, 34). It may be that the higher diversity of hydrogenotrophic methanogens means the pathway is robust, and capable of withstanding extreme conditions and shocks, or that hydrogen is a more easily accessible substrate for uptake.

**Medium-sized granules are the most active.** While the communities in small granules were the most diverse and, perhaps, the most susceptible to environmental change (based on NRI calculations), medium-sized granules were found to have the highest rates of methane production, potentially marking them as the most important size, or life stage, for AD system management. There are several possible physicochemical explanations for this phenomenon that were not measured in this study, such as observations on porosity, substrate diffusion, and specific surface area. However, it seems that the dynamics of the active community structure can give some insights into which taxa were significantly more active in these medium-sized (fractions E to G) granules. From the 56 discriminant genera, 28 individual genera were found to be upregulated in the medium-sized granules when supplied with a specific substrate. Statistically, those could have contributed to the methanogenic activity measured in the SMA assays. While many of the discriminants were fermenters and, therefore, higher up the AD hierarchy than we predicted, several interesting organisms were determined to be involved. For example, when provided with $H_2/CO_2$, *Methanobacterium* and *Methanolinea*, identified as discriminants, were found to be abundant in the active

**FIG 3** Legend (Continued)
Diversity was calculated using Shannon entropy (a to d) and rarefied richness (e to h) to minimal row sums of 53,117, 63,229, 74,100, and 101,511 for acetate, hydrogen, propionate, and butyrate, respectively. The shaded regions (using LOESS smoothing) track the diversity shifts of the substrate-supplied (pink) and no-added-substrate control (blue) samples. Lines connect two categories where the differences were significant (ANOVA): *, $P < 0.05$; **, $P < 0.01$; or ***, $P < 0.001$.

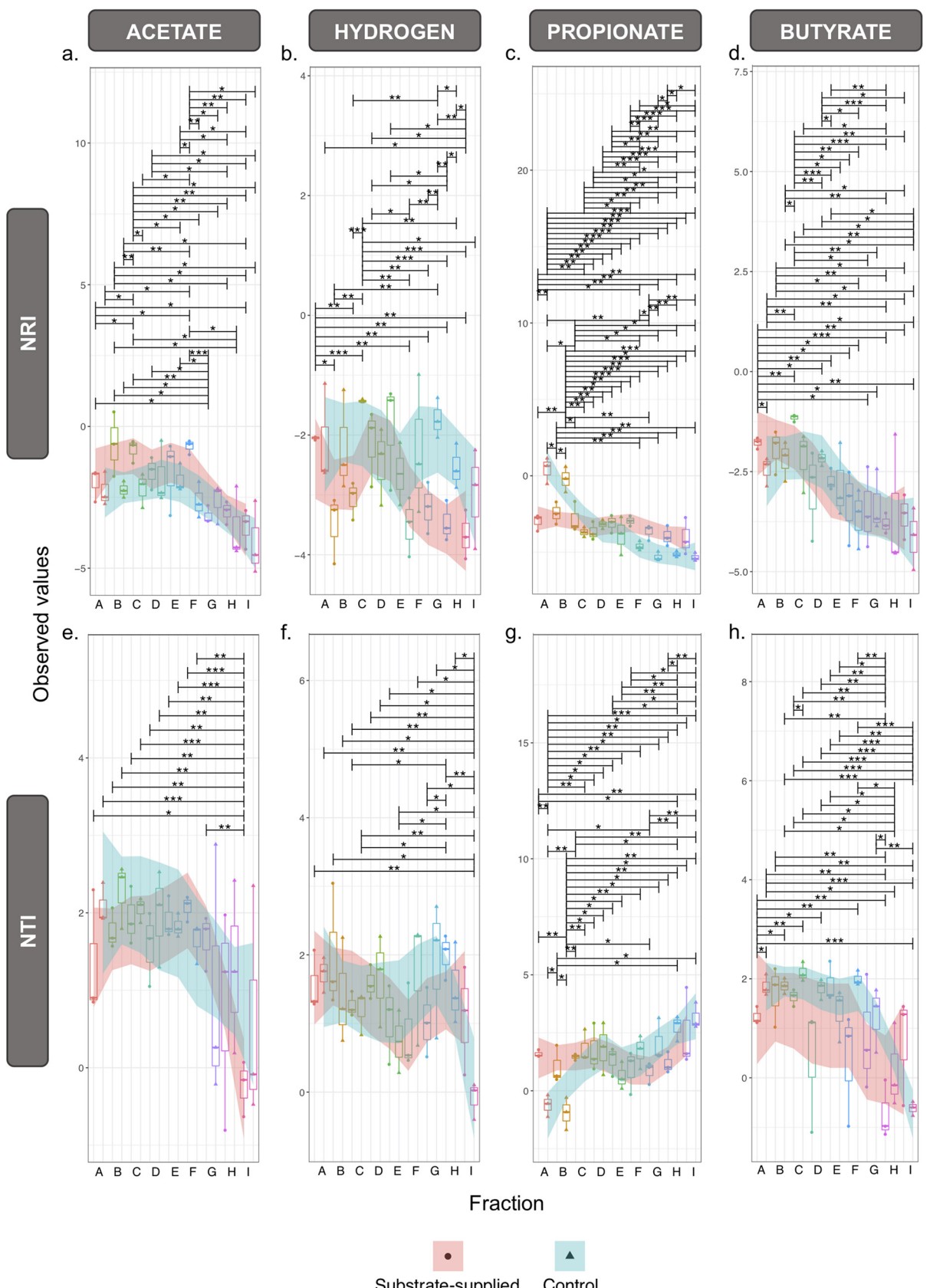

**FIG 4** Environmental filtering according to phylogenetic variances in rRNA (cDNA) profiles in samples ($n = 3$) from across each of the nine size fractions, A to I, supplied with specific substrates (acetate, $H_2/CO_2$, propionate, and butyrate) and the respective no-added-substrate controls,

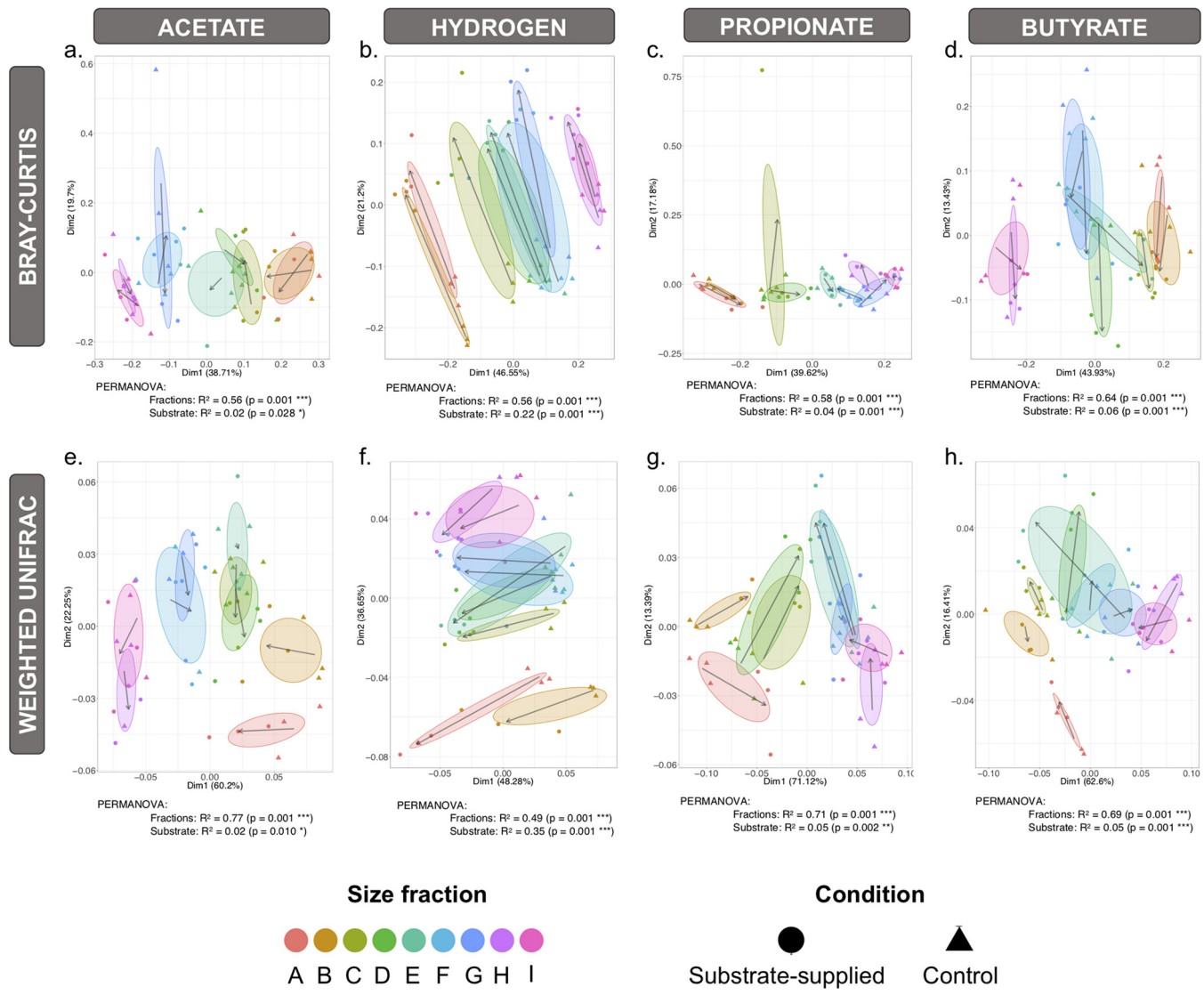

**FIG 5** Principal coordinate analysis (PCoA) using Bray-Curtis dissimilarity (a to d) and weighted UniFrac (e to h), where each point corresponds to the active community structure of one sample, either substrate supplied (acetate, hydrogen, propionate, or butyrate) or the respective no-added-substrate control. Size fractions are indicated by color, the ellipses are drawn at 95% confidence intervals for all samples from each size (including controls), and arrows mark the direction of change in the active community structure from mean ordination of control to the mean ordination of substrate supplied for each fraction, with the length indicating the amount of change. PERMANOVA (distances between groups) suggests more significant differences for size fractions than the substrate used. Note that the spread and elongation of the ellipse represent the significant shift in active community with respect to the substrate.

community of the medium-sized granules and were likely the main contributors to hydrogenotrophic methanogenesis.

## MATERIALS AND METHODS

**Source and fractionation of biomass.** Anaerobic sludge granules were sourced from a full-scale, mesophilic upflow anaerobic sludge bed (UASB) bioreactor treating potato-processing wastewater in the Netherlands. The sludge was size separated into nine size fractions (A to I) by passing the sludge through a range of stainless steel sieves. The volumetric contribution of each size fraction to the collective sludge sample was measured after allowing the granules to settle for 1 h.

**FIG 4** Legend (Continued)
calculated using the phylogenetic tree with presence/absence or abundance showing the net relatedness index (NRI) (a to d) and the nearest taxon index (NTI) (e to h) and where the shaded regions (using LOESS smoothing) track the phylogenetic alpha diversity shifts of the substrate-supplied (pink) and no-added-substrate control (blue) samples. Lines connect two categories where the differences were significant (ANOVA): *, $P < 0.05$; **, $P < 0.01$; ***, $P < 0.001$.

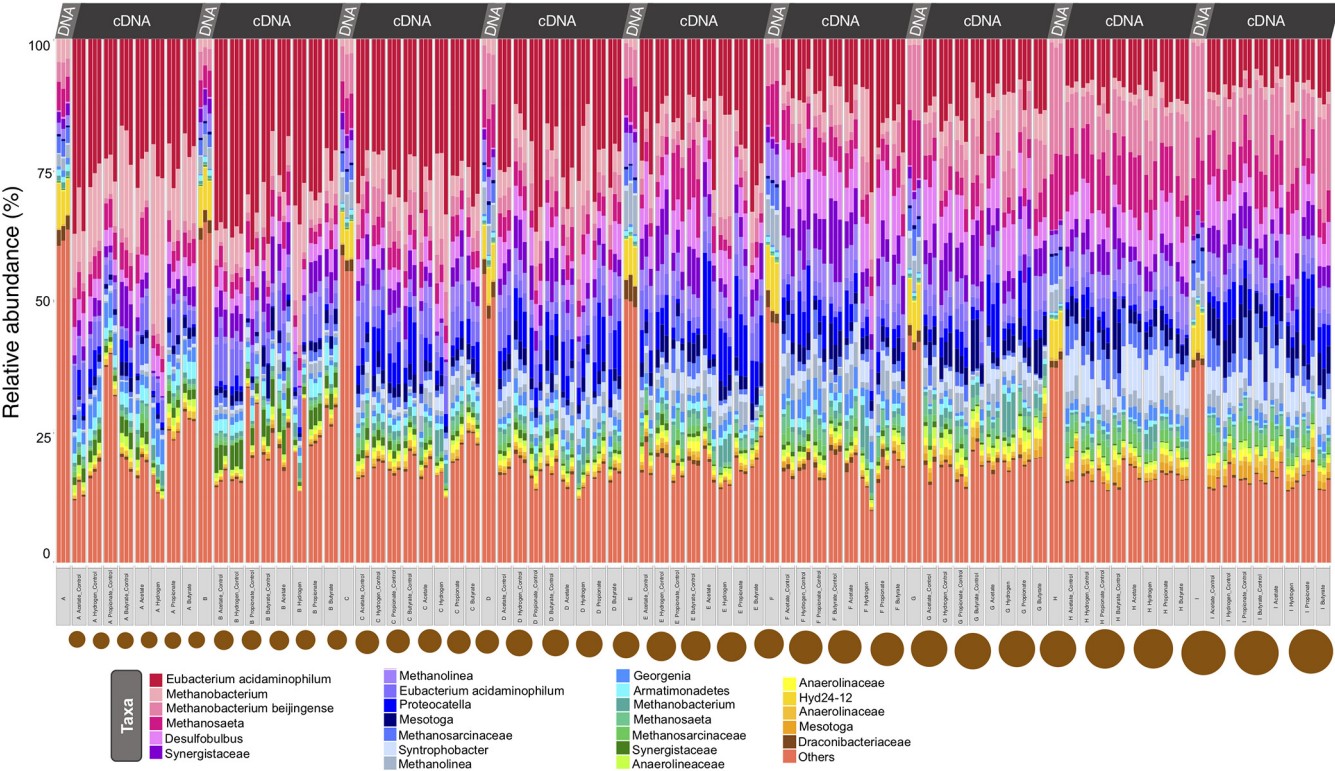

**FIG 6** Microbial diversity, and community structure, according to variances in rRNA profiles, in samples ($n = 3$) from across each of the nine size fractions, A to I, where community structure is based on relative abundance of the top 25 most abundant OTUs from T0 DNA from each size fraction and cDNA from each condition (substrate supplied or the respective no-added-substrate control), where "others" refers to all OTUs not included in the top 25.

**SMA and biomass sampling.** SMA assays were set up, in triplicate, to determine the maximum rate of methane production in biomass from each of the nine size fractions, A to I, against each of acetate, propionate, butyrate, and $H_2/CO_2$ and including no-added-substrate (or $N_2/CO_2$ in the case of negative controls for $H_2/CO_2$) controls ($n = 162$) by following previously described protocols (20, 21). Biogas production was monitored daily using a handheld pressure transducer (Centrepoint Electronics, Inc., Galway, Ireland). Separately, and once the SMA assays were concluded, each different incubation condition was set up again, but this time with a no-added-substrate control incubation for each condition for each of the size fractions ($n = 72$) to facilitate sacrificial biomass sampling at one time point during the exponential phase of biogas production. Biomass was immediately flash-frozen in liquid $N_2$ and stored at $-80°C$, while the liquid samples were stored for volatile fatty acid (VFA) analysis.

**VFA analysis.** VFA concentration profiles were determined for acetic, propionic, butyric, isobutyric, valeric, isovaleric, caproic, and isocaproic acids in the samples using a gas chromatograph equipped with a Combi PAL autosampler system (Varian, Inc., Walnut Creek, CA) and a 25-m-long CP-WAX 58 (FFAP) capillary column (SGE Analytical Sciences) with an internal diameter of 0.32 mm and 0.2-$\mu$m film thickness. Liquid samples in 1-ml aliquots were injected into the front injector (port type 1177) at 250°C. Helium gas carried the sample at a flow rate of 1 ml min$^{-1}$ under a constant flow and pressure of 0.1 lb/in². The oven temperature was 60°C for the first 0.1 min, 100°C for 1.17 min, and 200°C for 12.97 min. VFA were detected using a flame ionization detector (FID) at 300°C and identified and quantified by assigning chromatographic retention times against a calibration curve of known VFA and known concentrations. 2-Ethylbutyric acid was used as the internal standard.

**DNA/RNA coextraction, cDNA synthesis, and amplicon sequencing.** Total community DNA/RNA were coextracted from each sample (i.e., T0 biomass from each of the nine size fractions, in triplicate, along with triplicate samples from each of 72 time point samples; total of 243 samples) by following procedures described previously (35). For each sample investigated, a mass of 0.1 g wet sludge was transferred to respective sterile tubes in triplicate. Nucleic acids were extracted on ice by following the DNA/RNA coextraction method (35), which was based on bead beating in 5% (wt/vol) cetyltrimethyl-ammonium bromide (CTAB) extraction buffer, followed by phenol-chloroform extraction. The quality of nucleic acids was assessed using a NanoDrop spectrophotometer (Thermo Fisher Scientific, Waltham, MA, USA), and concentrations were determined using a Qubit fluorometer (Invitrogen, Carlsbad, CA, USA). Nucleic acids were stored at $-80°C$.

Samples were subsequently defrosted on ice, and RNA purification was achieved by applying a DNase treatment using a TurboDNase kit (AMBION-Invitrogen, Carlsbad, CA, USA) and following the manufacturer's recommendations. cDNA then was synthesized using a Superscript IV reverse transcriptase kit (Thermo Fisher, Waltham, MA, USA) by following the manufacturer's recommendations. Samples

were cooled on ice and cDNA concentrations determined using a Qubit (Invitrogen, Carlsbad, CA, USA). cDNA was stored at −20°C. 16S rRNA sequences from both DNA and cDNA were amplified using the universal bacterial and archaeal primer 515F and reverse primer 806R (36). The amplicon library of short inserts was sequenced on the Illumina MiSeq platform.

**Bioinformatics and statistical analysis.** Abundance tables were generated by constructing OTUs. Statistical analyses were performed in R using the combined data generated from the bioinformatics as well as metadata associated with the study. An OTU table was generated for this study by matching the original barcoded reads against clean OTUs (a total of 3,831 OTUs for $n = 246$ samples) at 97% similarity (a proxy for species-level separation). Alpha diversity analyses included the calculation of Shannon entropies and rarefied richness. Further multivariate integration (MINT) algorithms identified studywise discriminants, and additional details are available in the supplemental material.

**Data availability.** The sequencing data from this study are available in the European Nucleotide Archive under the study accession numbers PRJEB28212 (DNA samples) and PRJEB29752 (cDNA samples). The physicochemical data sets and scripts used during the current study are available from the corresponding author(s) on reasonable request.

## SUPPLEMENTAL MATERIAL

Supplemental material is available online only.

**TEXT S1**, DOCX file, 0.03 MB.
**FIG S1**, TIF file, 2.5 MB.
**FIG S2**, TIF file, 2 MB.
**FIG S3**, TIF file, 2.2 MB.

## ACKNOWLEDGMENTS

We thank NVP Energy for supplying anaerobic granules for the study.

C.Q. was supported by MRC fellowship MR/M50161X/1 as part of the CLoud Infrastructure for Microbial Genomics (CLIMB) consortium MR/L015080/1. U.Z.I. was supported by NERC IRF NE/L011956/1. A.C.T., S.O.S., S.M., and G.C. were supported by a European Research Council starting grant (3C-BIOTECH 261330) and by a Science Foundation Ireland Career Development Award (17/CDA/4658) to G.C. A.C.T. was further supported by a Thomas Crawford Hayes bursary from NUI Galway and a Short-Term Scientific Mission grant through the EU COST Action 1302. None of the funding bodies supporting this research had any role in the design of the study, in the collection, analysis, or interpretation of data, or in writing the manuscript.

We have no competing interests to declare.

A.C.T. and G.C. designed the study. A.C.T. performed all of the physicochemical characterizations, methanogenic activity assays, and wet molecular biology procedures, with assistance from S.O.S. and S.M. A.C.T. prepared the sequencing libraries. U.Z.I. wrote the scripts for data analysis, which was conducted by A.C.T. C.Q. contributed to the application of ecological theory. Results were interpreted by A.C.T., C.Q., U.Z.I., and G.C. A.C.T. drafted the paper, and C.Q., U.Z.I., and G.C. revised the document. All authors approve the paper and agree to accountability for the work therein.

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
