## [Reviewer comments · mSystems]

SIZE SHAPES THE ACTIVE MICROBIOME OF METHANOGENIC GRANULES, CORROBORATING A BIOFILM LIFE-CYCLE

Anna Trego, Sarah O'Sullivan, Christopher Quince, Simon Mills, Umer Ijaz, and Gavin Collins

Corresponding Author(s): Gavin Collins, National University of Ireland, Galway

Review Timeline:

Submission Date:	April 13, 2020
Editorial Decision:	June 11, 2020
Revision Received:	August 21, 2020
Accepted:	September 15, 2020

Editor: Paul Wilmes

Reviewer(s): Disclosure of reviewer identity is with reference to reviewer comments included in decision letter(s). The following individuals involved in review of your submission have agreed to reveal their identity: Magdalena Calusinska (Reviewer #1)

Transaction Report:

DOI: <https://doi.org/10.1128/mSystems.00323-20>

June 11, 2020

Dr. Gavin Collins
National University of Ireland, Galway
Microbiology
School of Natural Sciences
University Road
Galway, Galway H91TK33
Ireland

Re: mSystems00323-20 (SIZE, MORE THAN SUBSTRATE, SHAPES ACTIVE MICROBIOME OF METHANOGENIC GRANULES, CORROBORATING A BIOFILM LIFE-CYCLE)

Dear Dr. Gavin Collins:

Below you will find the comments of the reviewers.

To submit your modified manuscript, log onto the eJP submission site at <https://msystems.msubmit.net/cgi-bin/main.plex>. If you cannot remember your password, click the "Can't remember your password?" link and follow the instructions on the screen. Go to Author Tasks and click the appropriate manuscript title to begin the resubmission process. The information that you entered when you first submitted the paper will be displayed. Please update the information as necessary. Provide (1) point-by-point responses to the issues raised by the reviewers as file type "Response to Reviewers," not in your cover letter, and (2) a PDF file that indicates the changes from the original submission (by highlighting or underlining the changes) as file type "Marked Up Manuscript - For Review Only."

Due to the SARS-CoV-2 pandemic, our typical 60 day deadline for revisions will not be applied. I hope that you will be able to submit a revised manuscript soon, but want to reassure you that the journal will be flexible in terms of timing, particularly if experimental revisions are needed. When you are ready to resubmit, please know that our staff and Editors are working remotely and handling submissions without delay. If you do not wish to modify the manuscript and prefer to submit it to another journal, please notify me of your decision immediately so that the manuscript may be formally withdrawn from consideration by mSystems.

To avoid unnecessary delay in publication should your modified manuscript be accepted, it is important that all elements you upload meet the technical requirements for production. I strongly recommend that you check your digital images using the Rapid Inspector tool at <http://rapidinspector.cadmus.com/RapidInspector/zmw/>.

Sincerely,

Paul Wilmes

Editor, mSystems

Journals Department
Reviewer comments:

Reviewer #1 (Comments for the Author):

The study by Trego et al. describes the active microbiome of anaerobic granules at the different stages of their formation, here differentiated by their size. The structure of the active community in granules incubated with different substrates is defined with the use of the high-throughput 16S rRNA amplicon sequencing, which is a commonly used approach in molecular ecology. It is combined with the analysis of VFAs, here defined as targeted metabolomics and methane production. The selected bioinformatics tools are adequate for this kind of study.

Although my general impression is very positive, I miss some crucial information about the experimental design. For how long the experiment was carried out? Did the community had time to adapt to the fed substrate? I don't exactly understand if the purpose was to first adapt the community and then to characterise the active part of the microbiome? What were the characteristics of the source sample? If the original community was feed with mono substrate (I understood it was potato-processing wastewater), the structure/composition of granules was specifically adapted to it, and perhaps it would have needed an adaptation period before assessing an influence of the tested substrates on the active community. I follow the idea of the authors, and based on the presented results, indeed, it would look like the size more than the substrate shapes the activity. But, if the microbiome structure is not well adapted, perhaps the picture of the active community that we see is not completely correct, simply because the specific microbes are not there (or not in enough quantity to be detected) to answer to the test condition. That is why we might have an impression that rather size than the substrate shapes the active part of the microbiome. Could you please comment on this?

Another point, partially raised by the authors in the discussion section, is whether the structure of the individual granules would allow the tested substrates to "penetrate" inside the granule to have an impact of the active community? I could understand than the outer part of the granule is formed by fermentative bacteria, which in my mind could be less affected by tested VFAs. For example, hydrogen was shown to have a stronger effect, while it can also more freely reach the centre of the

granule. Why only simple VFAs were tested and not more complex substrates? Do you think that in the latter case the effect of the substrate on the active community would have been more pronounced?

The other minor comments/ questions are as follow:

1. With the manuscript structure containing Methods at the end of the publication, it would be beneficial to the reader to include a small description of the study design at the beginning of the Results section. There is Fig.1 that contains a graphic representation of the study design, but unless one goes to the methods, it is not completely clear how the whole experiment was designed.
2. Based on the Fig.6 it looks like the DNA was also sequenced, while it is not indicated in the Methods section. At least I understood that only RNA part was sequenced.
3. Figure 3 gives the rarefied richness, but nowhere in the manuscript is it indicated to what number of reads were the samples normalised. It is only written " to minimum library size". But it will make a difference if this size was rather 2 000 or 20 000 reads.
4. Based on my experience the universal primers underestimate the diversity of archaea. What was the reasoning of choosing this specific primer pair?
5. The used SILVA database is quite outdated. There are at least three or four more recent releases. Why was this database used for taxonomic annotation?
6. For how long were the SMA assays conducted? Why only one time point was selected for analysis?
7. Doing methanogenic assays in closed serum bottles (which I understand was the case here, please correct me if not true) raises the question of high gas partial pressure in the headspace of the bottle. Do you think it might have an impact of the active community structure and the fact that the hydrogenotrophic methanogenesis was prevailing under all conditions? Does DET takes place in granules?
8. Line 236.." methanogenic archaea (the phylum Euryarchaeota) dominated the core of the active community." Written this way, one might have an impression that methanogenic archaea are in the core "centre" of the granule, which is true, but which was not the meaning here, and it was neither determined here.
9. There are some typos, like double coma "," or ",," , e.g. lines 68, ...please revise.

Reviewer #2 (Comments for the Author):

This study assesses the structure and function of methanogenic granules that are separated based on size and then subjected to various substrates commonly encountered during AD in wastewater treatment. They measure methane production as a sign of activity and analyze 16S rRNA gene amplicon data from both the total consortia and active fraction (cDNA). This is a generally well-written and clearly presented paper, I have the following comments to help improve clarity.

- At first, I was perplexed that the authors repeatedly used the term "active" populations in the context of using 16S rRNA gene amplicon analysis. However, as you get deep into the results section MS, you finally realize that the authors have analyzed amplicon analysis from cDNA (which is of course makes using this terminology valid). I would suggest making this clearer to the reader at the early stages.
- Figure 1 is excellent! One of best and clearest depictions of granules I have come across!
- I found the high number of very large and dense figure excessive and feel that some of these could have been moved to the supplementary material. For example, just present one type of beta diversity metric in Fig. 5

- Line 226. «to suggest that diversity converges over time, with biofilm age, toward a core, active microbiome.» This type of "core" statement should be backed up by citation of a figure or a p-value. As it is now it is difficult to see what data supports this apart from some manual inspection of the PCA plots in Fig. 5
- Line 276: "None directly generates methane, those are likely syntrophic partners in the AD process." Sentence needs a re-write
- Line 287: what is meant by "and perhaps also most susceptible"
- Line 298: First mention of «fermenters» in the concluding paragraph, perhaps prudent to introduce this result further up in the MS.
- Several instances of erroneous punctuation throughout the text (misplaced , or .)

RESPONSE TO REVIEWERS

Responses in red

Reviewer #1 (Comments for the Author):

The study by Trego et al. describes the active microbiome of anaerobic granules at the different stages of their formation, here differentiated by their size. The structure of the active community in granules incubated with different substrates is defined with the use of the high-throughput 16S rRNA amplicon sequencing, which is a commonly used approach in molecular ecology. It is combined with the analysis of VFAs, here defined as targeted metabolomics and methane production. The selected bioinformatics tools are adequate for this kind of study. Although my general impression is very positive, I miss some crucial information about the experimental design. For how long the experiment was carried out? Did the community had time to adapt to the fed substrate? I don't exactly understand if the purpose was to first adapt the community and then to characterise the active part of the microbiome? What were the characteristics of the source sample? If the original community was feed with mono substrate (I understood it was potato-processing wastewater), the structure/composition of granules was specifically adapted to it, and perhaps it would have needed an adaptation period before assessing an influence of the tested substrates on the active community. I follow the idea of the authors, and based on the presented results, indeed, it would look like the size more than the substrate shapes the activity. But, if the microbiome structure is not well adapted, perhaps the picture of the active community that we see is not completely correct, simply because the specific microbes are not there (or not in enough quantity to be detected) to answer to the test condition. That is why we might have an impression that rather size than the substrate shapes the active part of the microbiome. Could you please comment on this?

Another point, partially raised by the authors in the discussion section, is whether the structure of the individual granules would allow the tested substrates to "penetrate" inside the granule to have an impact of the active community? I could understand than the outer part of the granule is formed by fermentative bacteria, which in my mind could be less affected by tested VFAs. For example, hydrogen was shown to have a stronger effect, while it can also more freely reach the centre of the granule. Why only simple VFAs were tested and not more complex substrates? Do you think that in the latter case the effect of the substrate on the active community would have been more pronounced?

The other minor comments/ questions are as follow:

1. With the manuscript structure containing Methods at the end of the publication, it would be beneficial to the reader to include a small description of the study design at the beginning of the Results section. There is Fig.1 that contains a graphic representation of the study design, but unless one goes to the methods, it is not completely clear how the whole experiment was designed.

Thank you for your comment, this has been added at the beginning of the results section.

2. Based on the Fig.6 it looks like the DNA was also sequenced, while it is not indicated in the Methods section. At least I understood that only RNA part was sequenced.

Thank you this has been clarified in the materials and methods section

3. Figure 3 gives the rarefied richness, but nowhere in the manuscript is it indicated to what number of reads were the samples normalised. It is only written "to minimum library size". But it will make a difference if this size was rather 2 000 or 20 000 reads.

The minimal row sums have been added to the figure legend.

4. Based on my experience the universal primers underestimate the diversity of archaea. What was the reasoning of choosing this specific primer pair?

The reasoning is based on a study done D'Amore 2016 **A comprehensive benchmarking study of protocols and sequencing platforms for 16S rRNA community profiling** in which various sequencing platforms, primers, library preparation techniques were compared using a synthetic community. They showed that the error rate of sequencing the V4 region using MiSeq yielded the best results. We recognize, however, that each choice introduces its own bias. WRT archaea, we observed high numbers of them (compared to other studies) so we don't think underestimating them is a major issue here.

5. The used SILVA database is quite outdated. There are at least three or four more recent releases. Why was this database used for taxonomic annotation?

This was the most current database during the time of analysis.

6. For how long were the SMA assays conducted? Why only one time point was selected for analysis?

The assays ran until 'completion' i.e. until biogas evolution plateaued. The way the assay works is that it measures a rate of methane production, when supplied with a specific substrate. It is not an enrichment assay, it simply measures a snapshot of the communities' methanogenic activity for a given set of conditions that the granules were exposed to. For each substrate-inoculum the time it takes to utilise the substrate will be different. The samples were taken during the exponential phase of biogas production (activity) measured using a handheld pressure transducer. This is all explained in the materials and methods section.

7. Doing methanogenic assays in closed serum bottles (which I understand was the

case here, please correct me if not true) raises the question of high gas partial pressure in the headspace of the bottle. Do you think it might have an impact of the active community structure and the fact that the hydrogenotrophic methanogenesis was prevailing under all conditions? Does DET takes place in granules?

It may be the case to some extent; however, this is the nature of most such tests we are aware of in most of the published reports in this field. As such, a great deal of the knowledge in this field viz. the activity of the constituent trophic groups has been derived from such set-ups. Our paper is applying a layer of molecular analysis over the assay to characterize the composition of the community apparently affected by the supply of specific substrates. We accept there are arguable flaws with (1) closed-bottle assays, and (2) the supply of specific substrates which is rather artificial relative to the operation of bioreactors. As to whether DET takes place in granules, this is a fascinating question and something we're interested in elsewhere but didn't focus on here.

8. Line 236.." methanogenic archaea (the phylum Euryarchaeota) dominated the core of the active community." Written this way, one might have an impression that methanogenic archaea are in the core "centre" of the granule, which is true, but which was not the meaning here, and it was neither determined here.

Yes, we understand the confusion here. "Core" in this case has been removed.

9. There are some typos, like double coma ",," or ",.", e.g. lines 68, ...please revise.

Thank you, these have been seen to.

Reviewer #2 (Comments for the Author):

This study assesses the structure and function of methanogenic granules that are separated based on size and then subjected to various substrates commonly encountered during AD in wastewater treatment. They measure methane production as a sign of activity and analyze 16S rRNA gene amplicon data from both the total consortia and active fraction (cDNA). This is a generally well-written and clearly presented paper, I have the following comments to help improve clarity.

- At first, I was perplexed that the authors repeatedly used the term "active" populations in the context of using 16S rRNA gene amplicon analysis. However, as you get deep into the results section MS, you finally realize that the authors have analyzed amplicon analysis from cDNA (which is of course makes using this terminology valid). I would suggest making this clearer to the reader at the early stages.

This has now been clarified in the introduction. Thank you for the comment.

- Figure 1 is excellent! One of best and clearest depictions of granules I have come across!

Glad you liked it!

- I found the high number of very large and dense figure excessive and feel that some of these could have been moved to the supplementary material. For example, just present one type of beta diversity metric in Fig. 5

The point in showing all of them was to show that we indeed used permanova in every combination possible and in each case size was the significant community driver. But I understand the business of the figure. It has been changed and the unifrac has been moved to supplemental materials.

- Line 226. «to suggest that diversity converges over time, with biofilm age, toward a core, active microbiome.» This type of "core" statement should be backed up by citation of a figure or a p-value. As it is now it is difficult to see what data supports this apart from some manual inspection of the PCA plots in Fig. 5

This statement has been amended in the manuscript.

- Line 276: "None directly generates methane, those are likely syntrophic partners in the AD process." Sentence needs a re-write

Thank you, this has been changed.

- Line 287: what is meant by "and perhaps also most susceptible"

This is based on the NRI calculations (now clarified in the manuscript) and also explained in more detail in the results section.

- Line 298: First mention of «fermenters» in the concluding paragraph, perhaps prudent to introduce this result further up in the MS.

This finding has now been added to the results section.

- Several instances of erroneous punctuation throughout the text (misplaced , or .)

These have been looked after. Thank you.

September 15, 2020

Dr. Gavin Collins
National University of Ireland, Galway
Microbiology
School of Natural Sciences
University Road
Galway, Galway H91TK33
Ireland

Re: mSystems00323-20R1 (SIZE SHAPES THE ACTIVE MICROBIOME OF METHANOGENIC GRANULES, CORROBORATING A BIOFILM LIFE-CYCLE)

Dear Dr. Gavin Collins:

Your manuscript has been accepted, and I am forwarding it to the ASM Journals Department for publication. For your reference, ASM Journals' address is given below. Before it can be scheduled for publication, your manuscript will be checked by the mSystems senior production editor, Ellie Ghatineh, to make sure that all elements meet the technical requirements for publication. She will contact you if anything needs to be revised before copyediting and production can begin. Otherwise, you will be notified when your proofs are ready to be viewed.

Sincerely,

Paul Wilmes
Editor, mSystems

Journals Department
File 1 (Supplemental Methods): Accept
Figure S2: Accept
Figure S1: Accept
Figure S3: Accept